# From Structure to Function: The Promise of PAMAM Dendrimers in Biomedical Applications

**DOI:** 10.3390/pharmaceutics17070927

**Published:** 2025-07-18

**Authors:** Said Alamos-Musre, Daniel Beltrán-Chacana, Juan Moyano, Valeria Márquez-Miranda, Yorley Duarte, Sebastián Miranda-Rojas, Yusser Olguín, Juan A. Fuentes, Danilo González-Nilo, María Carolina Otero

**Affiliations:** 1Escuela de Química y Farmacia, Facultad de Medicina, Universidad Andres Bello, Santiago 8370146, Chile; alamosmusresaid@gmail.com (S.A.-M.); beltranchd@gmail.com (D.B.-C.); joe.barish@gmail.com (J.M.); 2Center for Bioinformatics and Integrative Biology, Facultad de Ciencias de la Vida, Universidad Andres Bello, Santiago 8370146, Chile; valeria.marquez@unab.cl (V.M.-M.); yorley.duarte@unab.cl (Y.D.); danilo.gonzaleznilo@gmail.com (D.G.-N.); 3Department of Chemical Sciences, Faculty of Exact Sciences, Andres Bello University, Santiago 8370146, Chile; sebastian.miranda@unab.cl; 4Center for Theoretical & Computational Chemistry (CQT&C), Department of Chemical Sciences, Faculty of Exact Sciences, Andres Bello University, Santiago 8370146, Chile; 5Departamento de Química y Medio Ambiente, Universidad Técnica Federico Santa María, Avenida España 1680, Valparaíso 2390123, Chile; yusser.olguin@usm.cl Valparaiso; 6Centro Científico y Tecnológico de Valparaíso (CCTVal), Universidad Técnica Federico Santa María, Avenida España 1680, Valparaíso 2390123, Chile; 7Centro de Biotecnología, Universidad Técnica Federico Santa María, Avenida España 1680, Valparaíso 2390123, Chile; 8Laboratorio de Genética y Patogénesis Bacteriana, Universidad Andres Bello, Avda. República 330, Santiago 7591538, Chile; jfuentes@unab.cl; 9Centro de Investigación de Resiliencia a Pandemias, Facultad de Ciencias de la Vida e Instituto de Salud Pública, Avda. República 330, Santiago 7591538, Chile

**Keywords:** dendrimers, PAMAM, endocytosis, cytotoxicity

## Abstract

PAMAM dendrimers are distinguished by their capacity for functionalization, which enhances the properties of the compounds they transport, rendering them highly versatile nanoparticles with extensive applications in the biomedical domain, including drug, vaccine, and gene delivery. These dendrimers can be internalized into cells through various endocytic mechanisms, such as passive diffusion, clathrin-mediated endocytosis, and caveolae-mediated endocytosis, allowing them to traverse the cytoplasm and reach intracellular targets, such as the mitochondria or nucleus. Despite the significant challenge posed by the cytotoxicity of these nanoparticles, which is contingent upon the dendrimer size, surface charge, and generation, numerous strategies have been documented to modify the dendrimer surface using polyethylene glycol and other chemical groups to temporarily mitigate their cytotoxic effects. The potential of PAMAM dendrimers in cancer therapy and other biomedical applications is substantial, owing to their ability to enhance bioavailability, pharmacokinetics, and pharmacodynamics of active ingredients within the body. This underscores the necessity for further investigation into the optimization of internalization pathways and cytotoxicity of these nanoparticles. This review offers a comprehensive synthesis of the current literature on the diverse cellular internalization pathways of PAMAM dendrimers and their cargo molecules, emphasizing the mechanisms of entry, intracellular trafficking, and factors influencing these processes.

## 1. Introduction

### 1.1. Nanotechnology and Biomedical Applications

Nanotechnology, a discipline focused on the creation of nanostructures with precisely defined dimensions and configurations through atomic and molecular manipulations [1], has garnered significant attention in the field of biomedical applications, encompassing disease prevention, diagnosis, and treatment [2,3]. Nanoparticles represent a diverse class of structures, including liposomes, polymeric micelles, ceramic and polymeric nanoparticles, virus-derived capsid nanoparticles, polyplexes, and dendrimers [2,4,5]. Among these, dendrimers have emerged as particularly promising due to their unique structural characteristics, which facilitate the selective and controlled delivery of drugs, vaccines, and genetic material [6,7]. Their versatility, high degree of biocompatibility, and ability to traverse biological barriers render them strong candidates for the development of targeted nanocarriers [8]. Furthermore, research indicates that dendrimers can surpass conventional nanocarriers by providing enhanced drug solubility, greater chemical stability, and improved targeting precision [2,6,9].

### 1.2. Properties and Applications of Dendrimers

Dendrimers are highly branched, monodisperse nanoscale macromolecules with well-defined, symmetric, and globular architectures. Their unique structure features a central core, repeated branching layers (generations), and a high density of surface functional groups, which can be controlled during synthesis [3]. This allows for tunable size, molecular weight, solubility (including water solubility), and surface chemistry, making dendrimers highly versatile materials. They possess internal cavities capable of encapsulating guest molecules, and their surfaces can be engineered to display multiple functionalities, such as targeting ligands or charged groups, enhancing biocompatibility, and reducing toxicity [5,6].

### 1.3. Polyamidoamine Dendrimers: Innovations in Drug Delivery and Targeted Therapy

Polyamidoamine (PAMAM) dendrimers, in particular, demonstrate enhanced drug solubility and stability relative to liposomes and polymeric nanoparticles [10]. These benefits are largely ascribed to their highly branched and monodisperse architecture, which augments the surface area-to-volume ratio and facilitates their interaction with bioactive compounds. The internal cavities within the dendritic structure permit the efficient encapsulation of a diverse array of therapeutic agents, thereby improving their bioavailability and enabling targeted release. In specific therapeutic contexts, such as cancer treatment, PAMAM dendrimers have shown reduced systemic toxicity and enhanced drug efficacy [9,11], underscoring their utility as nanomedical platforms.

Structurally, dendrimers are highly branched macromolecules consisting of three main components: A central core, internal cavities formed by repetitive branching layers, and peripheral functional groups [5,12]. These architectural elements define their physicochemical properties, including molecular size, solubility, surface charge, and hydrophobicity [3,5,9,12]. The branches extend symmetrically from the core, forming a well-defined three-dimensional morphology that can be observed at the nanoscale (Figure 1) [12]. The chemical composition of terminal groups can be strategically modified to optimize interactions with biological targets, thereby enhancing specificity, stability, and pharmacological performance [3].

Dendrimers can be synthesized using divergent or convergent methodologies, both of which enable precise control over the size, generation number, and surface functionality [13,14]. The first dendritic molecules were reported by Vögtle in 1978 [12], and the most widely studied families to date include PAMAM dendrimers developed by Tomalia et al. [3,7] and poly(propylene imine) (PPI) dendrimers introduced through Newkome’s arboreal synthesis strategy [5,15]. Among these, PAMAM dendrimers are commonly synthesized via a divergent approach that begins with a multifunctional core molecule, often ethylenediamine, ammonia, or cystamine, and proceeds through iterative cycles of Michael addition and amidation [16,17]. This process requires strict control of the stoichiometry, reaction time, and temperature, along with efficient purification to minimize structural defects and maintain monodispersity [18,19]. Despite challenges such as steric hindrance, intramolecular cyclization, and characterization complexity at higher generations, PAMAM dendrimers are commercially available up to generation ten [20,21].

The physicochemical behavior of dendrimers is closely influenced by their branching architecture and surface chemistry. The nature of the branches, such as amine, ester, or ether-based units, affects properties such as hydrophobicity, solubility, and molecular recognition. Aliphatic chains enhance flexibility and aqueous solubility, whereas aromatic moieties confer rigidity and thermal stability to the compounds. Additionally, peripheral groups can be modified to promote specific interactions with biomolecules, enabling targeted delivery and improving diagnostic performance [7]. The progressive addition of generations increases structural complexity and porosity, which is crucial for encapsulating therapeutic agents and modulating their release. This hierarchical branching also determines the mechanical flexibility and capacity for surface functionalization of dendrimers [5,15].

Cargo loading in dendrimers occurs through entrapment within internal cavities and interactions with external functional groups. Internal voids serve as molecular pockets that protect encapsulated compounds from enzymatic degradation and premature release [15]. In parallel, the extensive hydrogen-bonding network provided by the branched structure enables the stable, noncovalent association of bioactive agents, contributing to controlled release kinetics and optimized bioavailability [22]. The surface charge of dendrimers, which depends on both the core composition and terminal group identity, plays a decisive role in cellular uptake, biodistribution, and therapeutic performance [14]. Moreover, functionalization with charged groups can enhance electrostatic interactions with biomolecules, facilitating the adsorption or complexation of nucleic acids, proteins, or other charged compounds [3,5].

Dendrimers can be classified according to their core type, branching structure, and surface modification strategy, with each configuration conferring specific physicochemical and biomedical properties. Among the most thoroughly characterized dendrimers are PAMAM dendrimers, which contain an ethylenediamine core and amide peripheral groups and are widely applied in drug delivery, gene therapy, and imaging [23]. Poly(propylene imine) (PPI) dendrimers, the first dendrimers synthesized on a large scale for industrial purposes, have also shown promise as therapeutic carriers. Poly-L-lysine (PLL) dendrimers exhibit excellent biocompatibility and are particularly suited for biomedical and gene delivery applications [24]. Other dendrimer families, such as carbosilane dendrimers with carbon-silicon backbones, offer remarkable chemical stability and have been explored for nanotechnology and materials science applications [25]. Phosphorus dendrimers, characterized by a cationic core and highly functionalized surfaces, are effective for gene transfection and molecular imaging [5]. Peptide dendrimers, comprising either polypeptide-based cores or peripheral peptide shells, offer modularity for precision-targeted therapeutic interventions [26]. Tryptophan-rich peptide dendrimers (TRPDs) have shown selective cytotoxicity against tumor cells [5]. Janus dendrimers, which possess dual-faced architectures with spatially segregated functional domains, provide precise molecular recognition and interaction, enhancing their potential for targeted drug delivery, gene therapy, and advanced biomaterial design [27].

The internalization of dendritic nanoparticles into cells is a critical determinant of their therapeutic efficacy. Uptake occurs primarily through endocytic and phagocytic pathways, with internalization dynamics influenced by the nanoparticle size, surface chemistry, and cell type specificity. Notably, spherical nanoparticles tend to exhibit faster and more efficient cellular uptake than their non-spherical counterparts, directly impacting their bioavailability and pharmacokinetic behavior [26,28]. The time scale of internalization also varies widely, ranging from seconds to several minutes, depending on both the nanoparticle characteristics and cellular context [29]. This variability highlights the necessity of precise control over dendrimer morphology, charge, and surface functionalization to optimize cellular interactions, reduce systemic toxicity, and enhance therapeutic outcomes. A deeper understanding of these interactions is essential for the rational design of next-generation nanocarriers that combine high specificity and minimal off-target effects. In summary, dendrimers represent a unique class of nanostructures with promising applications in drug delivery and biomedical engineering owing to their tunable properties and capacity for targeted delivery. Among these, PAMAM dendrimers are notable for their enhanced solubility and stability, making them a central focus of ongoing research in nanomedicine.

## 2. Materials and Methods

A comprehensive literature search was performed using the NCBI–PubMed, Google Scholar, and Mendeley databases. The search terms included “PAMAM dendrimers,” “drug delivery,” “gene delivery,” “biomedical applications,” and “nanomedicine,” individually and in combination using Boolean operators. Specific applications (e.g., “PAMAM + cancer therapy,” “PAMAM + siRNA,” and “PAMAM + toxicity”) were also explored to ensure thematic coverage.

The inclusion window spanned 1985–2025, capturing both early foundational discoveries and recent advances. Only peer-reviewed journal articles, including original research, reviews, and meta-analyses, were considered. The exclusion criteria involved conference abstracts, non-peer-reviewed content, and papers not directly related to PAMAM-based nanostructures.

Following screening and manual curation for relevance, methodological rigor, and application specificity, 137 scientific references were selected. These references were analyzed with respect to dendrimer synthesis, structural characteristics, functional modifications, biodistribution, biocompatibility, and therapeutic efficacy. This synthesis provides a comprehensive overview of PAMAM dendrimers as promising nanoplatforms in translational nanomedicine.

### 2.1. PAMAM Dendrimers

PAMAM dendrimers are among the most extensively studied dendritic nanostructures for biomedical applications. These synthetic macromolecules are composed of ethylenediamine or ammonia cores, from which homogeneous, repetitive layers of amide and amine groups radiate outward. Their highly branched and symmetrical architecture forms a compact and monodisperse morphology that facilitates the encapsulation and targeted delivery of a broad range of therapeutic agents, thereby enhancing bioavailability and pharmacological efficacy. PAMAM dendrimers have demonstrated significant improvements in drug solubility and stability, surpassing other nanocarriers in several therapeutic contexts [24]. For instance, improved outcomes have been observed in the delivery of chemotherapeutic agents, such as doxorubicin and paclitaxel, when formulated with PAMAM dendrimers compared to conventional systems [25]. Moreover, the peripheral surfaces of these nanostructures can be functionalized with specific targeting ligands, enabling selective accumulation in diseased tissues, reducing off-target effects, and increasing therapeutic precision [26].

PAMAM dendrimers are classified based on their generation number, which corresponds to the number of concentric branching layers extending from the central core of the dendrimer. Full generations are characterized by terminal amino groups, whereas half-generations are characterized by terminal carboxyl groups (Figure 1) [7,27]. This incremental growth leads to spherical, hyperbranched nanostructures that mimic the size and shape of natural proteins, earning them the designation of “artificial proteins” owing to their biomimetic properties [27,30]. One of the most advantageous features of PAMAM dendrimers is their tunable size and functional group density, which allow for the simultaneous incorporation of multiple therapeutic agents into a single nanoparticle. This characteristic supports the development of combination therapies with enhanced pharmacokinetics and therapeutic synergy [12,27,28,29].

Roberts et al. described the synthesis and initial biological evaluation of PAMAM dendrimers using a commercial system called Starburst™ dendrimers. This system presents a highly controlled synthesis method for spherical macromolecules formed by repeated poly(amidoamine) units produced in successive generations, each with a well-defined size, molecular weight, and number of terminal amino groups. This method involves stepwise synthesis, yielding molecules with an initiator core and concentric layers of branching, allowing precise control over the dendrimer structure and surface functionality. Dendrimers of generations 3 (G3), 5 (G5), and 7 (G7) were evaluated both in vitro using lung fibroblast cells (V79) and in vivo using Swiss-Webster mice. Toxicity, immunogenicity, and biodistribution were also assessed. Notably, only G7 exhibited relevant biological complications at high concentrations. No immunogenicity was observed in any of the cases, suggesting a favorable immunological profile for such scaffolds in biomedical applications. Regarding biodistribution, G3 primarily accumulated in the kidney, while G5 and G7 showed a preference for the pancreas, and G7 exhibited extremely high urinary excretion [31].

These dendrimers have been widely investigated in various therapeutic applications. In oncology, PAMAM-based nanocarriers improve drug targeting and release profiles, reducing systemic toxicity and enhancing antitumor efficacy [8]. Functionalized derivatives have also shown potential in anti-inflammatory therapies by prolonging the drug half-life and amplifying bioactivity [32]. Their antimicrobial applications are noteworthy; PAMAM dendrimers can act synergistically with antimicrobial agents or exhibit intrinsic antimicrobial activity [33]. Furthermore, their defined structure and modifiable surface chemistry make them efficient non-viral vectors for gene delivery, positioning them as promising tools for genetic therapy [34]. The ability to fine-tune the physicochemical properties of PAMAM dendrimers through controlled surface functionalization is critical for optimizing their biomedical application. Computational approaches, including Multiscale Molecular Simulations, have become essential for guiding the rational design and structural optimization of these nanocarriers [35,36]. These in silico strategies facilitate the prediction of molecular behavior, significantly reducing the time, cost, and complexity of experimental development [7,12,22,37,38,39,40,41,42].

Surface modifications often aim to mimic biologically compatible architectures by following biomimetic design principles. Such strategies have expanded the functionality of PAMAM dendrimers into diagnostic platforms, including their use as MRI contrast agents and theranostic tools [43]. The surface charge of PAMAM dendrimers plays a pivotal role in their biological behavior, particularly in their cytotoxicity. Cationic variants, especially those terminated with primary amine groups (–NH_2_), exhibit higher toxicity owing to their strong electrostatic interactions with negatively charged cell membranes. These interactions disrupt membrane integrity, increase permeability, and trigger cell lysis [2,6]. Elevated production of reactive oxygen species (ROS) and mitochondrial damage have also been associated with cationic dendrimers, further exacerbating their cytotoxic effects [13]. In contrast, anionic (–COOH) and neutral (–OH) surface-modified PAMAM dendrimers demonstrate substantially lower toxicity and improved biocompatibility [4,14], making them more suitable for applications that require minimal cellular disruption. Compared to poly(propylene imine) (PPI) dendrimers, PAMAM dendrimers offer superior aqueous solubility and biocompatibility. PPI dendrimers tend to be highly hydrophobic, which limits their colloidal stability in physiological environments [42]. Moreover, PAMAM dendrimers exhibit greater flexibility for surface functionalization, which is a critical advantage for customizing interactions in drug delivery systems.

PEGylation, which involves the attachment of polyethylene glycol (PEG) chains to surface amine groups, is a widely used strategy to enhance the biocompatibility of PAMAM dendrimers. PEGylation effectively shields positive surface charges, mitigating nonspecific interactions and reducing cytotoxicity while maintaining sufficient cellular uptake [44]. This modification also enhances nanoparticle stability in physiological media and prolongs circulation time, which is particularly advantageous for targeted delivery. For example, Fant et al. (2010) [45] demonstrated that PEGylation and acetylation of PAMAM dendrimers significantly influence their biological performance as gene delivery vectors. These surface modifications were shown to reduce the intrinsic cytotoxicity of PAMAM dendrimers, leading to improved biocompatibility [41].

PEGylation further enables the neutralization of the surface charge, reducing interactions with plasma proteins and minimizing clearance by the mononuclear phagocyte system. This stealth effect improves immune evasion and bioavailability of the drug. Additionally, computational studies at the atomic level have shown that PEG chain length significantly affects the drug-loading capacity and release dynamics of PEGylated dendrimers. Importantly, PEGylation markedly reduces the cytotoxicity of cationic dendrimers by masking reactive amine groups [46]. Fant et al. (2010) [45] demonstrated that PEGylation can modulate DNA binding and gene delivery efficiency in a size-dependent manner. Although PEGylation reduces cytotoxicity and DNA condensation, it may also lower transfection efficiency, depending on the molecular weight of PEG, dendrimer generation, and degree of PEGylation [45].

Overall, PAMAM dendrimers represent a highly versatile nanoplatform for drug delivery, gene therapy, and diagnostic imaging applications. Their precisely engineered architecture, combined with tunable surface functionality and charge, provides fine control over the physicochemical and biological properties of these nanomaterials. The integration of computational modeling with biomimetic design principles continues to expand the capabilities of dendrimers, bringing them closer to clinical translation as next-generation nanomedicines.

### 2.2. Loading Mechanisms and Functionalization of Dendrimers for Drug Delivery

Chemical functionalization is a pivotal strategy that transforms dendrimers into biologically active nanoplatforms, enabling their adaptation for diverse biomedical applications [47]. By conjugating functional moieties either on the dendrimer surface or within their internal cavities, these nanostructures can be tailored for the targeted and simultaneous delivery of multiple therapeutic agents to specific sites. This structural adaptability enhances pharmacokinetics and bioavailability, positioning dendrimers as highly effective carriers for combination therapy [12,30,37,39,47].

The mechanisms by which dendrimers interact with and deliver therapeutic compounds are largely dictated by their internal architectures and surface chemistries. Transport can occur either through physical encapsulation within dendritic cavities or by association with the outer surface. The dendrimer generation plays a crucial role in this context, as higher-generation structures possess an increased number of peripheral functional groups and larger internal volumes, which directly influence the loading capacity and diversity of compounds that can be accommodated [3,9]. Within these functionalizations, we highlight eight types in particular.

Physical Encapsulation

Among the primary loading mechanisms, physical encapsulation is particularly suited for hydrophobic drugs with low aqueous solubilities. The internal cavities of dendrimers provide a favorable microenvironment for the entrapment of such molecules through noncovalent interactions, including van der Waals forces and hydrogen bonding. This method is strongly influenced by the surface charge of the dendrimer [3], with higher-generation PAMAM dendrimers offering enhanced loading capacity and extended-release profiles owing to their increased cavity volume [48,49,50].

2.Electrostatic Complexation

Electrostatic complexation is another widely employed strategy, particularly for charged therapeutic agents such as siRNA. In this mechanism, oppositely charged drug molecules interact with the functional groups on the dendrimer surface to form stable ionic complexes. For instance, the electrostatic interaction between the carboxyl groups of ibuprofen and the terminal amines of PAMAM dendrimers significantly increases the aqueous solubility and stability of the drug after complexation [3,51,52,53].

3.Covalent Conjugation

Covalent conjugation is the third approach, in which drugs are chemically bound to the dendrimer surface using stable linkers. This strategy enhances the structural integrity of the nanocarrier and provides finely tunable drug release kinetics. Linkers such as polyethylene glycol (PEG) are often employed to improve solubility and circulation time. Amide bonds formed in this manner ensure prolonged drug retention, whereas ester bonds allow for a more rapid release under physiological conditions [3,29,54,55].

4.Surface Modification and Targeting Ligands

The biological performance of dendrimers is profoundly influenced by the nature of their external functional groups, which govern recognition, binding, and internalization by target cells [56]. Consequently, a wide range of functionalization strategies has been developed to enhance the targeting specificity. For example, folic acid has been used to exploit the overexpression of folate receptors in cancerous and infected tissues. In studies by Ilyes Benchaala et al., folate-functionalized PAMAM dendrimers exhibited 3- to 4-fold higher accumulation in folate receptor-rich tissues compared to non-functionalized counterparts, supporting their utility in selective drug delivery [57]. Moreover, pretreatment with agents such as all-trans retinoic acid (ATRA) can increase folate receptor expression, further enhancing the targeting efficacy of folate-dendrimer conjugates carrying chemotherapeutics such as docetaxel [57]. Molecular dynamics simulations have demonstrated that PEG linkers, particularly PEG 3350, improve solubility and receptor-binding affinity without compromising the targeting functions of these conjugates [58].

5.Alternative Ligands and Stimuli-Responsive Systems

Other targeting ligands, including epidermal growth factor (EGF), have also been successfully conjugated to dendrimers for gene delivery to cancer cells that overexpress EGFR. In a study by Li et al., EGF-functionalized dendriplexes exhibited enhanced transfection efficiency, selective accumulation in EGFR-positive tumors, and minimal off-target toxicity, highlighting their potential as gene therapy platforms [59]. Recent advances have incorporated pH-responsive release mechanisms that exploit the acidic tumor microenvironment to trigger localized drug release, thus improving the therapeutic index while minimizing systemic exposure [60].

6.Multifunctionalization: Antibodies, Peptides, Aptamers, Vitamins, and Sugars

Dendrimers have also been functionalized with monoclonal antibodies, peptides (e.g., RGD or laminin-binding sequences), aptamers (such as MUC1 and AS1411), vitamins, and sugars such as glucosamine. These moieties enable the active targeting of tumor-specific receptors that are overexpressed in various malignancies, including breast cancer. For example, peptide-functionalized dendrimers have been used for the targeted delivery of gemcitabine to colorectal cancer cells [61], while aptamer-conjugated dendrimers have facilitated the selective delivery of epirubicin to breast and colon tumors [62]. Additionally, PEGylated dendrimers passively accumulate in tumor tissues through the enhanced permeability and retention (EPR) effect, further increasing their therapeutic potential [63]. These functionalization strategies collectively improve biocompatibility, reduce systemic toxicity, enable stimuli-responsive release, and promote tumor-specific accumulation, reinforcing dendrimers as robust platforms for targeted therapy and diagnostics.

7.Functionalized Dendrimers with Intrinsic Therapeutic Activity

In addition to drug delivery, certain functionalized dendrimers exhibit intrinsic therapeutic activities. Abd-El-Aziz et al. reported that dendrimers modified with ferrocenyl groups and quaternary ammonium or 2-mercaptobenzothiazole moieties displayed potent antimicrobial activity. These compounds were effective against multidrug-resistant pathogens, such as methicillin-resistant Staphylococcus aureus (MRSA), vancomycin-resistant Enterococcus faecium, Staphylococcus warneri, Pseudomonas aeruginosa, and Candida albicans, underscoring their potential as novel antimicrobial agents [64].

Further investigations have highlighted the capacity of third- to fifth-generation PAMAM dendrimers to inhibit or reverse amyloid fibril formation associated with Alzheimer’s disease (AD). These dendrimers can directly interact with fibrils, prevent peptide aggregation, and accelerate the disassembly processes. These findings suggest that dendrimers may function as standalone therapeutic agents independent of drug encapsulation [65].

8.Ligand-Mediated Targeting and Cellular Uptake

Ligand-mediated targeting remains one of the most effective approaches for directing dendrimers to specific tissues in vivo. Conjugation with biomolecules, such as vitamins, peptides, and antibodies, enables selective interactions with overexpressed receptors on target cells, thereby improving therapeutic outcomes while reducing off-target toxicity [66]. Dendrimers designed for receptor-mediated uptake typically enter cells via clathrin-dependent endocytosis, ensuring efficient internalization and intracellular delivery [67]. This versatility has enabled the development of dendrimer-based systems for a broad spectrum of applications, including cancer therapy, autoimmune disease management, and antimicrobial resistance. In gene therapy, dendrimers have demonstrated excellent potential as non-viral vectors capable of protecting nucleic acids and facilitating targeted delivery to correct genetic disorders [68].

In summary, the functionalization of PAMAM dendrimers plays a central role in modulating their physicochemical and biological properties. Their ability to encapsulate, electrostatically bind, and covalently conjugate therapeutic agents makes them valuable tools for advanced drug delivery, antimicrobial applications, gene therapy, and diagnostic imaging. Emerging strategies, including ligand-mediated targeting, pH-responsive systems, and biomimetic modifications, have significantly expanded their clinical potential. As the field progresses, the continued integration of computational modeling and rational design is expected to accelerate the translation of dendrimer-based nanomedicines into precise clinical interventions.

### 2.3. Cellular Internalization Mechanisms of PAMAM Dendrimers

A comprehensive understanding of the interaction between PAMAM dendrimers and cellular membranes and their internalization by cells is essential for optimizing their therapeutic performance in both extracellular and intracellular applications. In intracellular contexts, the successful translocation of dendrimers across the plasma membrane is critical for accessing target organelles or cytoplasmic compartments, directly influencing their bioactivity and therapeutic efficacy [69]. The mechanism of cellular entry not only determines the efficiency of intracellular delivery but also affects the stability and functional outcome of the dendrimer-drug complex. Currently, different methods of dendrimer internalization are used, varying in their strategy and performance (Figure 2).

PAMAM dendrimers can be internalized through two main pathways: active transport via endocytosis and passive diffusion across the plasma membrane [9,69,70,71,72]. Among the active mechanisms, endocytosis is the predominant route and is strongly influenced by factors such as cell type, membrane lipid composition, and physicochemical properties of the dendrimer [69,73,74,75,76]. This energy-dependent process involves the invagination of the plasma membrane to engulf extracellular material into vesicles that are then trafficked into the cytoplasm. Following internalization, intracellular trafficking pathways determine the fate of dendrimer cargo, directing it towards specific organelles or facilitating its release into the cytosol [69,73,74,77,78] (Table 1).

Clathrin-mediated endocytosis represents one of the most extensively characterized routes of cellular uptake and is considered the canonical pathway for nanoparticle internalization [83]. This process is initiated by the assembly of clathrin triskelions, which are recruited and organized by adaptor proteins, such as AP-2 and AP180, to form a polyhedral lattice at the cytoplasmic surface of the membrane [84]. Vesicle scission from the membrane is mediated by the GTPase dynamin, which assembles at the neck of the budding vesicle and facilitates its release into the cytoplasm upon GTP hydrolysis [69,84]. Once internalized, clathrin-coated vesicles rapidly shed their coat and fuse with early endosomes, proceeding through the endolysosomal pathway, where their contents may be directed to lysosomes for degradation [84].

Importantly, cationic PAMAM dendrimers can exploit the “proton sponge” effect to escape endosomal compartments before lysosomal degradation. This escape mechanism is driven by the protonation of uncharged amine groups within the dendrimer structure, resulting in osmotic swelling and disruption of the endosomal membranes. This property is particularly relevant for gene delivery applications, where the successful release of genetic material into the cytoplasm is essential for transfection [84,85]. Despite its utility, the efficiency of this mechanism is variable and remains under active investigation, especially in terms of balancing endosomal escape and cytotoxicity [41].

Another relevant internalization mechanism is caveolae-mediated endocytosis, a clathrin-independent process that is predominant in endothelial, adipocyte, fibroblast, and muscle cells. Caveolae are flask-shaped membrane invaginations enriched in cholesterol and sphingolipids, and their formation depends on caveolin-1, which oligomerizes and binds to membrane cholesterol to initiate vesicle formation [84]. A key advantage of this pathway is its ability to bypass lysosomal degradation, allowing for prolonged intracellular retention and enhanced cargo bioavailability. This mechanism is particularly favorable for delivering proteins and nucleic acids and appears to be preferentially employed by anionic dendrimers, underscoring the critical role of surface charge in determining the uptake pathway [73,86].

Macropinocytosis is an additional clathrin- and caveolae-independent endocytic mechanism. This non-selective process is initiated by actin-driven membrane ruffling, leading to the formation of large vesicles (macropinosomes) that engulf extracellular fluid and solutes [84,87,88]. It is often triggered by growth factor stimulation, which activates receptor tyrosine kinases and downstream signaling cascades, including the production of phosphatidylinositol-3,4,5-trisphosphate (PIP3), driving actin polymerization and membrane extension. Macropinocytosis is particularly relevant for dendrimer-based cancer therapies, as many tumor cells exhibit elevated macropinocytic activity, enhancing the uptake of nanoparticles designed for targeted treatment [89].

In addition to these energy-dependent mechanisms, certain dendrimers can cross the cell membrane via passive diffusion. This energy-independent process is driven by concentration gradients and does not require receptor engagement or vesicle formation. Studies have shown that PAMAM dendrimers of generations 4 (G4) and 6 (G6) can penetrate the plasma membrane following pretreatment with DL-buthionine-(S,R)-sulfoximine (BSO), a compound that increases membrane permeability [71,90,91]. Once inside the cytoplasm, dendrimers internalized through passive diffusion can exert antioxidant effects by scavenging reactive oxygen species (ROS), similar to small-molecule antioxidants such as N-acetylcysteine (NAC) and its amide derivative (NACA). This mechanism may reduce cytotoxicity and enhance therapeutic biocompatibility, making passive diffusion a promising strategy for developing dendrimer-based systems with minimal adverse effects [28,71,75].

Thus, PAMAM dendrimer internalization is a multifactorial process modulated by dendrimer generation, surface charge, and membrane composition. Clathrin- and caveolae-mediated endocytosis are the predominant energy-dependent pathways, whereas macropinocytosis is increasingly recognized as a significant mechanism, particularly in cancer cells. Passive diffusion, although less common, represents a complementary strategy that can enhance cytosolic delivery under specific conditions. A nuanced understanding of these internalization routes is essential for the rational design of dendrimer-based nanocarriers, enabling improved intracellular targeting, reduced degradation, and enhanced therapeutic performance. Continued research on the molecular determinants of dendrimer uptake will inform the development of next-generation nanomedicines for gene therapy, targeted drug delivery, and precision oncology.

### 2.4. Intracellular Trafficking of PAMAM Dendrimers

The internalization of PAMAM dendrimers into cells represents only the initial phase of their intracellular transport. The subsequent trafficking processes within the cellular environment are equally critical, if not more critical, in determining their biological efficacy. Intracellular trafficking is a highly regulated and dynamic process influenced by multiple factors, including dendrimer shape, surface charge, functional group composition, and specific phenotype of the target cell [85,92]. These variables introduce significant complexity and hinder the establishment of a universal set of structural criteria for optimal dendrimer performance in therapeutic applications.

Biological barriers, such as membrane-associated efflux systems, can further restrict intracellular accumulation, particularly in contexts characterized by drug resistance, such as multidrug-resistant cancer cells and antibiotic-resistant bacterial strains [93]. These efflux mechanisms actively transport foreign substances, including nanocarriers, out of the cytosol, thereby limiting the bioavailability of the therapeutic agent. A distinctive feature of PAMAM dendrimers is their structural tunability, which enables precise modulation of physicochemical parameters, such as size, morphology, surface potential, and chemical functionality. These modifications are pivotal in directing dendrimers to specific tissues or cell types, thereby facilitating the development of targeted delivery strategies. However, once internalized, dendrimers must overcome a secondary set of challenges to exert their intended effects on the target cells. These include enzymatic degradation, clearance by innate immune mechanisms, nonspecific interactions with cytoplasmic biomolecules, and the need to traverse intracellular membranes to reach their sites of action. Despite these hurdles, PAMAM dendrimers have shown considerable promise for intracellular delivery of nucleotide- and protein-based therapeutics. Their capacity to protect sensitive biomacromolecules from degradation and enable cytosolic release makes them attractive platforms for gene therapy applications, including RNA interference and gene silencing strategies targeting disease-related genes [93].

In addition to their role as carriers, PAMAM dendrimers have potential applications in diagnostic imaging. Their ability to form stable complexes with metal ions has been exploited in the development of magnetic resonance imaging (MRI) contrast agents, which contribute to enhanced signal intensity and improved spatial resolution [93]. This dual functionality (therapeutic and diagnostic) positions dendrimers as valuable components for the design of next generation theranostic nanoplatforms.

#### 2.4.1. Nucleus-Targeted PAMAM Dendrimers

The nucleus is a key intracellular target for therapeutic interventions, particularly in gene therapy, where the precise delivery of nucleic acids is essential for achieving clinical efficacy. PAMAM dendrimers have considerable potential in this domain because of their ability to form stable electrostatic complexes with DNA, thereby protecting the genetic material from nuclease-mediated degradation and facilitating cellular uptake [44]. The interaction between the phosphate groups of DNA and the cationic surface of PAMAM dendrimers results in compact, stable complexes that enhance transfection efficiency by condensing nucleic acids into structures amenable to nuclear transport [22,94].

Among non-viral gene delivery vectors, PAMAM dendrimers have been extensively investigated. Their surface-exposed amine groups allow for efficient DNA binding, a property that is further amplified in higher-generation dendrimers owing to the increased surface charge density [95,96]. Moreover, internal tertiary amines contribute to the “proton sponge” effect, which promotes endosomal escape, an essential step for cytoplasmic release and subsequent nuclear translocation [6]. These dendrimers typically exhibit low immunogenicity and can be chemically modified to improve their biocompatibility, targeting specificity, and cargo release [4]. However, this study had several limitations. Cytotoxicity is a prominent concern, particularly in higher-generation variants (e.g., G5 and above), where elevated amine content can lead to membrane destabilization and cell death [13]. In addition, the non-biodegradable nature of PAMAM dendrimers raises questions regarding their long-term accumulation and potential in vivo toxicity [14], whereas the complex, time-consuming synthesis of higher-generation dendrimers presents challenges for large-scale production [12]. Compared to other synthetic vectors, PAMAM dendrimers exhibit a unique balance of transfection efficiency and biocompatibility. Polyethyleneimine (PEI), often regarded as a benchmark for gene delivery, demonstrates high transfection rates but also exhibits significantly greater cytotoxicity [3]. In contrast, chitosan, a natural cationic polysaccharide, offers excellent biocompatibility and biodegradability but requires chemical modification to achieve comparable gene delivery efficacy [7]. Poly(β-amino esters) (PBAEs) are another alternative with inherent biodegradability and synthetic versatility; however, their performance is highly dependent on structural parameters [5].

Achieving an optimal balance between surface charge and particle size is essential for efficient nuclear targeting while minimizing cytotoxicity. Jia-Ying Yan et al. demonstrated that the conjugation of thymine residues to a fourth-generation PAMAM dendrimer (G4) reduced its overall surface charge, leading to diminished complex stability with DNA [97]. Nonetheless, this surface modification favored nuclear localization in cancer cells, suggesting a context-dependent effect. Once internalized, dendrimers must traverse the cytoplasm to reach the nucleus before releasing their cargo. Several strategies have been explored to enhance nuclear targeting efficiency. One approach involves modifying the dendrimer surface with nitrogenous base analogs to increase the affinity for the nucleus. Yan et al. reported that G4-thymine-functionalized dendrimers preferentially accumulated in the nuclei of H460 lung cancer cells, potentially due to increased thymine demand in rapidly dividing cells compared to non-cancerous HEK293 kidney cells [97].

Intracellular transport can also be enhanced by interactions with microtubule networks. Samuel D. Jativa et al. developed a dendrimer conjugated with a dynein-binding peptide (DBP), enabling active transport along microtubules via interaction with dynein light chain 8 [98]. This strategy facilitated efficient intracellular trafficking to the perinuclear regions. The size of the dendrimer is another critical parameter influencing nuclear import, given the size-selective nature of the nuclear pore complex (NPC). Guan-Hai Wang et al. engineered a second-generation PAMAM dendrimer equipped with an outer layer linked via disulfide bonds. Once internalized, intracellular glutathione cleaves disulfide linkages, reducing nanoparticle size and promoting nuclear entry [98].

Another widely applied strategy involves the use of nuclear localization signals (NLS). Ji Li et al. developed a PEGylated PAMAM dendrimer to evade rapid clearance by the mononuclear phagocyte system and extend systemic circulation time. The dendrimer was further functionalized with arginine-glycine-aspartate (RGD) peptides to enhance integrin-mediated cellular uptake. Upon internalization, intracellular reductive conditions cleave PEG chains, exposing an HMGB1-derived NLS peptide that facilitates nuclear translocation and improves transfection efficiency [99].

#### 2.4.2. Mitochondria-Targeted PAMAM Dendrimers

Mitochondria are central regulators of cellular metabolism and apoptosis, rendering them strategic targets for therapeutic interventions in cancer, metabolic disorders, and neurodegenerative diseases. Mitochondrial dysfunction has been implicated in the pathogenesis of numerous conditions, including Alzheimer’s disease, Parkinson’s disease, amyotrophic lateral sclerosis (ALS), and multiple sclerosis, where excessive oxidative stress and impaired mitochondrial dynamics contribute significantly to disease progression [100,101,102]. During oxidative phosphorylation, mitochondria generate reactive oxygen species (ROS) as byproducts, with approximately 1–2% of the consumed oxygen giving rise to ROS production. These species can damage essential biomolecules, such as membrane phospholipids, proteins, and DNA, leading to cellular dysfunction and cell death [103]. Furthermore, the aberrant regulation of mitochondrial apoptotic pathways plays a critical role in oncogenesis, highlighting the relevance of mitochondrial targeting in anticancer strategies [40,104]. The unique mitochondrial membrane potential, characterized by a high negative charge, facilitates the selective accumulation of lipophilic cationic compounds in mitochondria. Among these, triphenylphosphonium (TPP) has been widely used because of its delocalized positive charge, which drives its electrostatic attraction to mitochondria [105]. Beyond its targeting ability, TPP exhibits intrinsic anticancer properties, likely associated with ROS overproduction in metabolically active cancer cells, making it a promising candidate for mitochondrial-directed therapy [106,107].

Several studies have functionalized PAMAM dendrimers with TPP to enhance mitochondrial localization. For instance, Liang et al. developed a TPP-conjugated PAMAM dendrimer for photodynamic therapy, achieving improved mitochondrial targeting and therapeutic efficacy [108]. Similarly, Maghsoudnia et al. employed a fifth-generation TPP-PAMAM (G5-TPP) construct to deliver let-7b microRNA, a regulator of mitochondrial respiratory complexes that is frequently downregulated in cancer. Their results confirmed efficient mitochondrial accumulation and apoptosis induction in A549 lung carcinoma cells [109]. The potential of mitochondrial-targeted dendrimers extends to the delivery of small-molecule drugs to the mitochondria. Curcumin, a known ROS inducer and cytochrome c releaser, was incorporated into a PAMAM-TPP platform by Sadeghizadeh et al., resulting in enhanced cytotoxicity against hepatocellular carcinoma cells while preserving selectivity for cancer cells over normal counterparts [110].

Beyond oncology, mitochondrial-targeted PAMAM dendrimers have demonstrated their value in neuroprotective applications. Conventional antioxidants, such as N-acetylcysteine (NAC), often fail to reach mitochondrial compartments because of their limited permeability. To overcome this limitation, Sharma et al. synthesized a PAMAM-TPP-NAC conjugate (TDN), which was effectively localized in mitochondria and mitigated hydrogen peroxide-induced oxidative damage in microglial cells, highlighting its therapeutic potential in neurodegenerative disease models [102]. Overall, successful intracellular trafficking of PAMAM dendrimers is fundamental to their functions. While nuclear-targeted dendrimers have gained attention for gene delivery, mitochondrial-targeted systems offer considerable promise for treating disorders characterized by mitochondrial dysfunctions. By leveraging the mitochondrial membrane potential, these dendrimers enable selective mitochondrial accumulation, thereby enhancing the therapeutic precision. As research progresses, the rational design of dendrimer-based nanocarriers will continue to advance, contributing significantly to the development of next-generation precision medicines.

However, the biosafety profile of PAMAM dendrimers remains a critical concern. Despite their utility, dendrimers can interact with cellular components in undesirable ways, potentially leading to cytotoxicity. These include membrane destabilization and increased ROS production, both of which are known to cause DNA damage and apoptosis. Therefore, elucidating the toxicity mechanisms of dendrimers is essential for predicting their in vivo behaviors and mitigating their adverse effects. Experimental studies have shown that cytotoxicity is influenced by multiple parameters, including dendrimer concentration, exposure duration, generation number, and surface charge, which are primarily governed by the nature of the terminal functional groups (Figure 3) [28,30,111,112]. Understanding and controlling these factors is paramount to ensure the safe and effective application of dendrimer-based therapies.

The cytotoxicity associated with PAMAM dendrimers arises through multiple mechanisms, most notably influenced by the nature and density of the surface functional groups, which confer distinct electrostatic properties to the dendrimers. These terminal groups may impart a cationic (e.g., amine-terminated), anionic (e.g., carboxylate-terminated), or neutral (e.g., hydroxyl-terminated) character to the dendrimer surface. Among these, cationic dendrimers exhibit the highest cytotoxicity, primarily because of their strong electrostatic interactions with negatively charged cellular membranes [111,112,113,114]. Specifically, primary amine-terminated PAMAM dendrimers have been shown to disrupt membrane integrity by forming nanopores, leading to the leakage of intracellular contents and subsequent cell death. This membrane-disruptive effect intensifies with increasing dendrimer generation due to the greater number of surface amines [111,112,113,114]. In addition to membrane disruption, surface charge and dendrimer size are closely associated with intracellular stress responses, including reactive oxygen species (ROS) generation, lysosomal pathway activation, apoptosis induction, and DNA damage [30,112]. Interestingly, this cytotoxic potential has also been exploited for antimicrobial applications, with dendrimers demonstrating intrinsic bactericidal activity attributed to their membrane-interacting properties [86,89].

At the mitochondrial level, cationic PAMAM dendrimers accumulate because of the high negative membrane potential, promoting localized ROS production that triggers the apoptotic cascade [111]. Studies using HaCaT keratinocytes have revealed that G4 to G6 cationic PAMAM dendrimers induce significant mitochondrial stress, manifested by increased ROS generation, lysosomal activation, DNA fragmentation, and apoptotic cell death [114]. Furthermore, dendrimers have been shown to interfere with critical intracellular signaling pathways. For instance, increased dendrimer generation has been associated with the downregulation of transcriptional regulators, such as EGR1, TFPI2, and IGFBP3; inhibition of the AKT/TSC2/mTOR axis; and activation of ERK1/2 signaling—events that can lead to apoptosis or autophagy [115,116,117,118]. Notably, even neutral (–OH) functionalized dendrimers have been shown to produce ROS and affect mitochondrial function, although organelles such as chloroplasts and photosynthetic membranes remain largely unaffected [112,114,119]. Another important mechanism of dendrimer-induced cytotoxicity is the indirect depletion of essential extracellular proteins. Certain dendrimers can nonspecifically adsorb serum proteins, reducing the availability of key biomolecules necessary for cellular function. This phenomenon also exhibits generation dependence, with higher-generation dendrimers exhibiting more pronounced effects [111,119]. Additionally, hemolytic activity has been reported, particularly with cationic PAMAM dendrimers, which induce rapid red blood cell lysis compared to their anionic or neutral counterparts. The hemolytic potential is both concentration- and generation-dependent, with higher-generation amine-terminated dendrimers producing significantly more hemolysis [30,113,120,121].

Understanding the toxicological profile of PAMAM dendrimers is critical for their clinical application in humans. One of the most effective strategies for mitigating their cytotoxic effects is surface modification, which aims to modulate the net charge of the nanostructure. Among the most extensively used approaches is the conjugation of biocompatible and non-immunogenic moieties, such as carbohydrates, anionic groups, amino acids, or peptides, to the dendrimer surface, which reduces interactions with cellular membranes [37,111,122]. PEGylation has emerged as a prominent therapeutic strategy. The incorporation of polyethylene glycol (PEG) chains not only decreases the surface positive charge, thereby reducing cytotoxicity, but also improves solubility and circulation time. Moreover, the terminal hydroxyl groups of PEG facilitate further functionalization for specific biomedical applications [37,122].

### 2.5. Harnessing PAMAM Dendrimer Cytotoxicity for Biomedical Applications

The inherent versatility and structural stability of PAMAM dendrimers have positioned them as promising nanocarriers for transporting therapeutic agents, with the potential to mitigate the adverse effects associated with conventional treatments, such as chemotherapeutic regimens [111,123,124,125]. Although cytotoxicity is often viewed as a limitation for the clinical translation of dendrimers, this property may also be strategically exploited in biomedical applications. Selectivity is essential in the treatment of infectious diseases and cancer; therapeutic systems must effectively eliminate pathogenic organisms or malignant cells while sparing healthy tissue. Achieving such specificity requires careful optimization of dendrimer dosage, incorporation of targeting ligands, and structural modifications to enhance cellular uptake and retention within diseased tissues [112]. Dendrimers have been extensively explored for their oncological applications. One example is the conjugation of paclitaxel to a generation 4 polyamidoamine (PAMAM) dendrimer via a bio-labile linker. This design enabled the selective activation of the prodrug by cathepsin B, an enzyme overexpressed in certain cancer cells, resulting in increased cytotoxicity in breast cancer cell lines exhibiting moderate to high cathepsin B activity compared to unconjugated drugs [126]. Similarly, targeted delivery has been achieved by conjugating trastuzumab, a monoclonal antibody against HER2, to a docetaxel-loaded polyamidoamine (PAMAM) dendrimer. This construct demonstrated enhanced antiproliferative activity and selectivity towards HER2-positive MDA-MB-453 breast cancer cells compared with HER2-negative MDA-MB-231 cells. Enhanced cellular internalization and apoptosis induction were observed in the HER2-positive model, indicating successful receptor-mediated targeting [47].

In addition to improving therapeutic efficacy, dendrimer-based systems have shown promise in modulating pharmacokinetic behaviors. Studies have reported significant improvements in the circulation time and biodistribution of docetaxel when it is administered as a dendrimer conjugate. This targeted delivery approach is expected to reduce systemic toxicity by limiting off-target effects and enabling site-specific drug accumulation, which is not afforded by the free drug [127,128]. More broadly, dendrimers conjugated or loaded with antineoplastic agents have consistently demonstrated the capacity to enhance drug bioavailability, prolong systemic half-life, and reduce dose-limiting toxicities [3,38,47,112]. The ability to fine-tune these nanostructures to balance cytotoxicity and specificity underscores their growing relevance in the development of next-generation cancer therapies.

### 2.6. Navigating Biosecurity Challenges of PAMAM Dendrimers for Biomedical Applications

PAMAM dendrimers, renowned for their precise architecture and functional versatility, have emerged as promising platforms in biomedicine, particularly for the targeted delivery of therapeutic and diagnostic agents. However, their clinical development has been significantly hindered by biosecurity concerns, as their inherent toxicity, especially in unmodified cationic forms, remains a critical obstacle that risks confining their use to the experimental domain [28,111,128,129,130]. The primary source of toxicity in PAMAM dendrimers lies in their dense surface of positively charged terminal amine groups, which foster strong electrostatic interactions with negatively charged cell membranes, leading to membrane disruption, nanopore formation, compromised cellular integrity, and ultimately, cell lysis [128,129]. Such toxicity has been clearly documented in vitro, where cationic PAMAM dendrimers have been shown to induce oxidative stress, alterations in mitochondrial membrane potential, activation of apoptotic and necrotic pathways, autophagic processes, and changes in gene expression that could have unforeseen implications in therapeutic contexts [28,111]. The extent of toxicity is closely linked to dendrimer generation, with higher generations, such as G4 and G5, exhibiting more pronounced effects due to the increased surface density of functional groups, which intensifies interactions with cellular and extracellular components [111,130].

At the systemic level, animal models have shown that PAMAM dendrimers can cause toxic effects, including hepatocellular vacuolization, hemolysis, and hematological changes such as reduced erythrocyte counts, hemoglobin, and hematocrit. These effects tend to intensify with higher doses and prolonged use [128,129]. Furthermore, the biodistribution and elimination of PAMAM dendrimers are influenced by particle size and surface chemistry. Lower-generation dendrimers are generally efficiently eliminated through renal excretion, while higher-generation dendrimers tend to accumulate in organs like the liver and spleen, increasing the risk of organ-specific toxicity [129,130]. Another crucial factor is interspecies variability and physiological differences affected by environmental factors, such as seasonality, which lead to varying toxicological outcomes and complicate the extrapolation of preclinical data to human clinical scenarios [129]. To address these limitations, several strategies have been proposed and developed to enhance the biosecurity of PAMAM dendrimers without compromising their effectiveness as targeted drug delivery systems. Among these strategies, PEGylation has been effective in reducing electrostatic interactions with cellular membranes, thereby decreasing cytotoxicity and extending systemic circulation time. However, repeated administration of PEGylated PAMAM dendrimers has been linked to accelerated blood clearance and immunological reactions, including hypersensitivity and the formation of anti-PEG antibodies, highlighting the need to explore alternatives such as polyoxazolines or polyvinylpyrrolidone [130]. Similarly, surface modifications that replace cationic amine groups with neutral or anionic groups, such as carboxyl, hydroxyl, or pyrrolidone residues, have significantly reduced cellular toxicity and hemolytic activity. However, these changes may reduce the dendrimers’ ability to efficiently interact with target biomolecules, potentially compromising their functionality in applications like gene delivery [111,129].

The use of lower-generation dendrimers, such as G3 or below, has been proposed as an effective strategy to mitigate toxicity. This is due to their reduced surface functional group density and more efficient renal elimination, which in turn decreases tissue accumulation and the risk of systemic toxicity [111]. However, this approach is limited by lower payload capacities and potentially reduced structural stability, both of which are crucial in therapeutic applications that require high functional group density for transporting therapeutic agents or genetic material [111,130]. In parallel, the development of biodegradable dendrimers, constructed from cores and branches that are susceptible to enzymatic or hydrolytic degradation—such as those based on citric acid or polyesters—has emerged as a promising strategy to reduce tissue persistence and long-term toxicity. Nonetheless, these designs are still in the early experimental stages and require thorough characterization before potential clinical translation [128]. The fundamental challenge in developing PAMAM dendrimers for biomedical applications lies in balancing the reduction of toxicity while preserving the functional properties that make them valuable platforms. The very features responsible for their high affinity for interacting with nucleic acids, proteins, or contrast agents are also implicated in their intrinsic toxicity [28,129]. As Labieniec-Watala and Watala (2015) [129] noted, the toxicity of cationic dendrimers remains a significant obstacle in their preclinical and clinical development. Although modified dendrimers exhibit more favorable pharmacological profiles, the central challenge persists in designing structures that harmonize safety and functionality without compromising therapeutic efficacy. Future advancements in this field will depend on integrating rational molecular design, innovative surface modification strategies, and rigorous toxicological assessments to ensure the biosecurity of nanomaterials in complex clinical settings [28,111,128,129,130].

### 2.7. Applications and Translational Challenges of Surface-Modified PAMAM Dendrimers in Biomedical Therapeutics

PAMAM dendrimers have emerged as promising nanocarriers for vaccine delivery, particularly in the context of DNA and peptide vaccines, owing to their highly branched architecture and chemically modifiable surface groups [131]. Functionalization of dendrimers with targeting moieties, such as peptides that bind to MHC class II molecules on antigen-presenting cells (APCs), has demonstrated enhanced delivery specificity and improved transfection efficiency, leading to robust immune activation. Notably, this strategy has been associated with the generation of high-affinity T cell responses and tumor rejection in vivo [132]. In a related approach, lysine-functionalized dendrimers (PAMAM-Lys) were employed to deliver a DNA vaccine against Schistosoma japonicum, yielding significant improvements in protective efficacy by reducing worm and liver egg burdens compared to unmodified DNA delivery [133]. Further advancements include mannosylated polyamidoamine (PAMAM) dendrimers that target mannose receptors expressed on dendritic cells, thereby enhancing antigen internalization and presentation. These constructs have been shown to stimulate robust antigen-specific CD4^+^ and CD8^+^ T cell responses, induce dendritic cell maturation, and delay tumor progression in murine models [134]. Collectively, these examples illustrate how surface-engineered PAMAM dendrimers can be tailored to optimize vaccine delivery by increasing cellular uptake, protecting antigens from degradation, enabling sustained release, and functioning as intrinsic immune adjuvants.

Despite their preclinical success, several physicochemical characteristics of dendrimers influence their applicability in humans. Parameters such as size (generation), surface chemistry, and structural rigidity critically affect the biodistribution, pharmacokinetics, and systemic clearance [82]. Higher-generation dendrimers exhibit greater rigidity and size, which reduces their deformability and limits their ability to traverse biological barriers such as the glomerular filtration membrane. Consequently, their renal clearance is often slower than that of linear polymers with similar molecular weights (MWs). Surface modifications, including PEGylation, have been employed to improve pharmacokinetics, reduce toxicity, and extend systemic circulation time, thus enhancing the clinical utility of nanocarriers [135].

In summary, the surface chemistry, size, and structural attributes of PAMAM dendrimers play decisive roles in determining their in vivo behavior and clinical performance [135]. Optimizing these properties is essential for improving therapeutic efficacy, minimizing toxicity, and facilitating the successful clinical application of dendrimer-based delivery systems in drug delivery. PAMAM dendrimers exhibit exceptional potential in a wide range of biomedical applications owing to their tunable physicochemical characteristics, which enable effective membrane interaction and intracellular access. Their intrinsic structural versatility supports the encapsulation or conjugation of multiple therapeutic agents, ultimately improving the pharmacokinetics, pharmacodynamics, and bioavailability of diverse active compounds [9,15,27,28]. However, their clinical translation remains challenging because of unresolved concerns regarding their internalization pathways and cytotoxicity profiles. A deeper understanding of the mechanisms governing cellular entry and the specific intracellular effects they induce is essential to advance their therapeutic utility. Future research should explore how the modification of internalization pathways can mitigate or eliminate the cytotoxic responses associated with PAMAM dendrimers. This approach diverges from conventional strategies, which primarily focus on surface modification, and presents a novel direction for improving the biosafety and specificity of these nanomaterials.

## 3. Conclusions

PAMAM dendrimers have significant potential in various biomedical applications owing to their controllable physicochemical properties that facilitate interactions with cell membranes and enable cellular entry. The ability to functionalize these nanoparticles by incorporating additional molecules enhances the pharmacokinetics, pharmacodynamics, and bioavailability of various active ingredients [9,15,27,28]. However, as noted, while dendrimers can generate biological actions upon cellular entry, they are not without associated toxicity. However, several concerns remain regarding the mechanisms of entry and processes leading to cytotoxicity.

Further research is essential to elucidate the biological effects of PAMAM dendrimers, focusing on optimizing internalization pathways and assessing their cytotoxic potential. Investigating these pathways could provide new strategies for mitigating the associated cytotoxicity by altering the cellular entry routes. This novel approach offers significant advantages beyond the current focus on modifying dendrimer surface properties, ultimately advancing their clinical applications in targeted drug delivery and therapy.

## Figures and Tables

**Figure 1 pharmaceutics-17-00927-f001:**
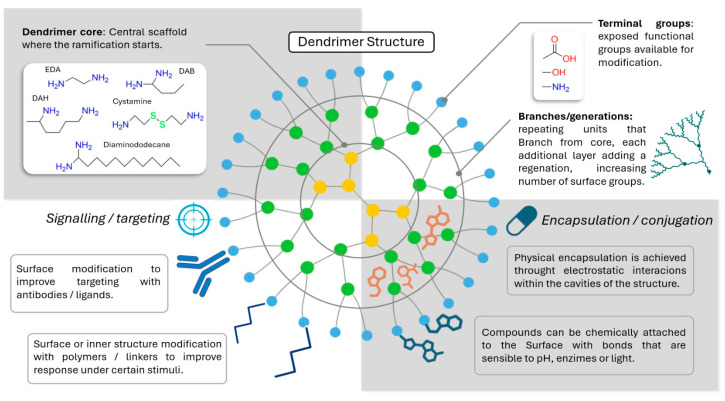
Schematic representation of the hierarchical and branched architecture of a dendrimer molecule, illustrating its core, repetitive branching units, and terminal functional groups.

**Figure 2 pharmaceutics-17-00927-f002:**
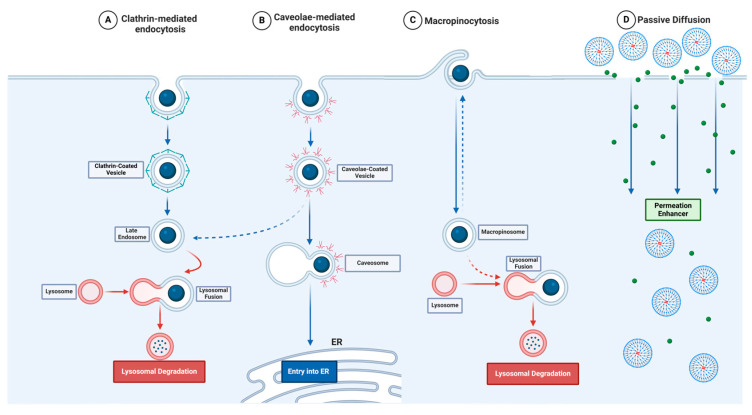
Illustration of different mechanisms of endocytosis of PAMAM-associated nanoparticles. (**A**) Clathrin-mediated endocytosis, showing the assembly of clathrin to the plasma membrane and the formation of the clathrin-coated vesicle with the dendrimer inside, followed by the late endosome, which then fuses with the lysosome for degradation. (**B**) Caveolae-mediated endocytosis, showing the formation of a caveolar vesicle with the dendrimer inside, showing the ability of this pathway to bypass the lysosomes by not heading towards the route leading to the late endosome and instead heading towards the endoplasmic reticulum by fusing with the organelle called the caveosome. (**C**) Macropinocytosis: We observed the formation of membrane ruffles that engulfed the dendrimer, forming an intracellular vacuole called a macropinosome, which then fused with lysosomes for degradation. (**D**) Passive diffusion: We observed the passive diffusion of PAMAM using a membrane permeation enhancer, causing nanopores that allow its diffusion through the membrane.

**Figure 3 pharmaceutics-17-00927-f003:**
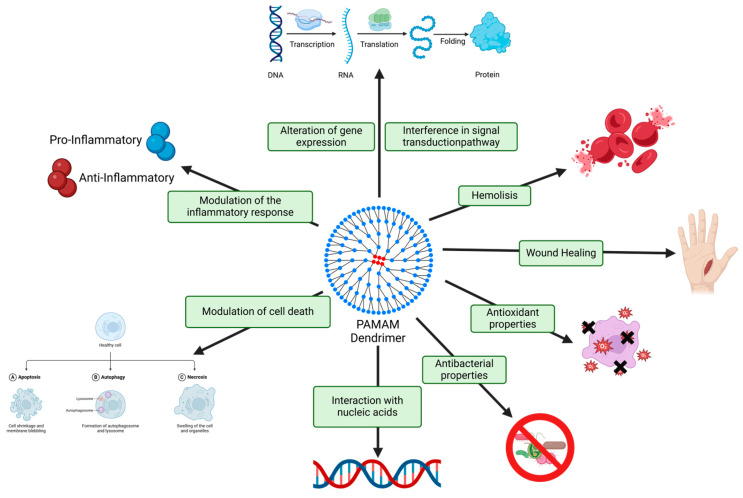
Illustration of the toxic effects and innate biological actions of PAMAM dendrimers. Toxic effects of PAMAM dendrimers include cell death and hemolysis. Innate biological actions are also observed, for example, acting as an antimicrobial, modulator of the inflammatory response, interfering in signal transduction pathways, among others [15].

**Table 1 pharmaceutics-17-00927-t001:** Comparison of the Effects of Different Cell Types and Membrane Compositions on PAMAM Dendrimer Internalization.

Cell Type	Surface Functionalization	Observed Internalization Pathway(s)	Key Observations	Reference
Microglia/Macrophages	Hydroxyl (G4-OH)	Likely macropinocytosis and phagocytosis	Rapid and extensive uptake; >80% in 3 h; >95% in 6 h	[79]
Astrocytes	Hydroxyl (G4-OH)	Minimal internalization	Only ~8.5% uptake after 24 h	[79]
Hippocampal Neurons	Unmodified	Clathrin-mediated endocytosis	Efficient internalization of unmodified PAMAM	[80]
Hippocampal Neurons	Folic acid (PFO)	Clathrin + caveolin-mediated endocytosis	Enhanced uptake via dual mechanisms	[80]
Hippocampal Neurons	PEG (50%)/Acrylate (30%)	Minimal uptake	Surface shielding inhibits internalization	[80]
HT-29 (Colon Cancer)	Unmodified/Propranolol-G3	Caveolin + macropinocytosis	Dual pathway uptake; propranolol improves cytotoxicity	[81]
HT-29 (Colon Cancer)	Lauryl-modified G3	Clathrin + caveolin + macropinocytosis	Surface hydrophobicity enhances multi-pathway internalization	[81]
Generic (various cells)	Anionic dendrimers	Caveolin-mediated endocytosis	Internalization correlates with charge and membrane interaction	[82]
Generic (various cells)	Neutral/Cationic dendrimers	Clathrin-independent endocytosis	Uptake efficiency varies with surface charge and membrane composition	[82]

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
