# Peer review of "From Structure to Function: The Promise of PAMAM Dendrimers in Biomedical Applications"

_pharmaceutics, 2025, doi:10.3390/pharmaceutics17070927_

Round 1

Reviewer 1 Report (Previous Reviewer 2)

Comments and Suggestions for Authors

This review manuscript is valuable, but some revisions are needed.

1. The logical order of the introduction section is too confusing. After introducing the topic of dendrimers, the authors should have introduced the properties and applications of dendrimers before introducing PAMAM, which belongs to a type of dendrimers. In addition, the author should add a paragraph summarizing the entire text.

2. In the section of “Loading Mechanisms and Functionalization of Dendrimers for Drug Delivery “, the authors should set up different subheadings to list the different functionalizations, otherwise it will confuse the reader.

3. Authors should add a reference to Figure 2 in the text.

4.The “Table: Comparison of the Effects of Different Cell Types and Membrane Compositions on PAMAM Dendrimer Internalization” should be Table1. Also, it should be mentioned in the suitable at the appropriate place in the text.

5. In the section of “Intracellular Trafficking of PAMAM Dendrimers”, the author mentioned the “However, the biosafety profile of PAMAM dendrimers remains a critical consideration.”. This section should be entitled in a separate paragraph and not merged under the above heading.

6.In the section of “Applications of Surface-Modified PAMAM Dendrimers in Vaccine Delivery and Immunogenicity Enhancement”, the authors describe the application of PAMAM in vaccine delivery only in the first two paragraphs. But the next few paragraphs deviate from the subject of this section.

7. The author has divided the entire article into too many small paragraphs and suggests synthesizing the relevant content into one large paragraph.

8. The title of the article is “Navigating the cellular labyrinth: the journey of PAMAM dendrimers”, but the content of PAMAM cell internalization is relatively small. It is recommended to change the title or adjust the main content of the article.

9. All abbreviations need to be explained only on the first mention, not on every mention.

Author Response

Thank you for your thoughtful feedback regarding our manuscript. We appreciate your comments and the opportunity to clarify some points related to our findings. New titles and paragraphs are marked in red.

Revisor 1:

  1. The logical order of the introduction section is too confusing. After introducing the topic of dendrimers, the authors should have introduced the properties and applications of dendrimers before introducing PAMAM, which belongs to a type of dendrimers. In addition, the author should add a paragraph summarizing the entire text.

R: We have reorganized the introduction to first present the properties and applications of dendrimers before introducing Polyamidoamine (PAMAM) dendrimers. Additionally, we included a summary paragraph at the end of the introduction to encapsulate the key points.

  1. In the section of “Loading Mechanisms and Functionalization of Dendrimers for Drug Delivery “, the authors should set up different subheadings to list the different functionalizations, otherwise it will confuse the reader.

R: the suggested changes were made. Line: 251, 258, 264, 271, 284, 292, 304 and 317.

  1. Authors should add a reference to Figure 2 in the text.

R: Figure 2 was created by us based on different sources of information.

  1. The “Table: Comparison of the Effects of Different Cell Types and Membrane Compositions on PAMAM Dendrimer Internalization” should be Table1. Also, it should be mentioned in the suitable at the appropriate place in the text.

R: The table number is now Table 1 and added as part of the text, specifically on line number 365.

  1. In the section of “Intracellular Trafficking of PAMAM Dendrimers”, the author mentioned the “However, the biosafety profile of PAMAM dendrimers remains a critical consideration.”. This section should be entitled in a separate paragraph and not merged under the above heading.

R: We thank the reviewer for the suggestion. As recommended, we have moved the discussion of biosafety into a separate section now titled “Navigating Biosecurity Challenges of PAMAM Dendrimers for Biomedical Applications.” This improves clarity and distinguishes biosafety considerations from intracellular trafficking.

  1. In the section of “Applications of Surface-Modified PAMAM Dendrimers in Vaccine Delivery and Immunogenicity Enhancement”, the authors describe the application of PAMAM in vaccine delivery only in the first two paragraphs. But the next few paragraphs deviate from the subject of this section.

R: The name of the section was changed to avoid confusion.

  1. The author has divided the entire article into too many small paragraphs and suggests synthesizing the relevant content into one large paragraph.

R: We appreciate the reviewer’s observation. In response, we have revised the manuscript by consolidating shorter paragraphs into longer, cohesive sections to improve flow and readability, while maintaining clarity of the scientific content.

  1. The title of the article is “Navigating the cellular labyrinth: the journey of PAMAM dendrimers”, but the content of PAMAM cell internalization is relatively small. It is recommended to change the title or adjust the main content of the article.

R: Title has change for: “From Structure to Function: The Promise of PAMAM Dendrimers in Biomedical Applications”.

  1. All abbreviations need to be explained only on the first mention, not on every mention.

R: All abbreviations are mentioned only once and when appropriate.

Reviewer 2 Report (New Reviewer)

Comments and Suggestions for Authors

The review by María Carolina Otero et al. is devoted to a rather interesting topic, namely the migration of PAMAM-type dendrimers into the cell.

The review is well structured and easy to read. The main aspects of the interaction of PAMAM dendrimers with cell components are considered, namely:

1) Mechanisms of penetration (passive diffusion and endocytosis)

2) intracellular transport, using the example of substrate delivery to mitochondria

And the main aspects of the use of PAMAM dendrimers, primarily the creation of targeted delivery systems for drugs, vaccines and genetic material.

The authors naturally could not help but discuss the toxicity of PAMAM dendrimers, and cited several basic and most convenient strategies for reducing toxicity, such as:

1) PEGylation

2) Surface functionalization

3) Ligandization and  vectors conjugation.

In my opinion, a supromolecular approach is missing, namely association with surfactants and the introduction of PAMAM dendrimers as components of extracellular vesicles or synthetic micelles. But from the context of the review, it is clear that the authors focused on the most popular approaches.

The strengths of the review include the fact that all stages of the "life" of dendrimers in the cell, the creation of pH-sensitive systems, and kinetic significance are covered.

Disadvantages.

References 28 and 41 are duplicated.

28. Zhu W, Okollie B, Bhujwalla ZM, Artemov D. PAMAM dendrimer-based contrast agents for MR imaging of Her-2/ neu
receptors by a three-step pretargeting approach. Magn Reson Med. 2008 Apr 26;59(4):679–85.

41. Zhu W, Okollie B, Bhujwalla ZM, Artemov D. PAMAM dendrimer-based contrast agents for MR imaging of Her-2/ neu
receptors by a three-step pretargeting approach. Magn Reson Med. 2008 Apr 26;59(4):679–85.

In my opinion, it would be a great omission not to mention the work of Jeanette C. Roberts, especially Preliminary biological evaluation of polyamidoamine (PAMAM) StarburstTM dendrimers (https://onlinelibrary.wiley.com/doi/10.1002/(SICI)1097-4636(199601)30:1%3C53::AID-JBM8%3E3.0.CO;2-Q).

Link 63, which is the second one (conjugates carrying chemotherapeutics
such as docetaxel (63).) leads to the wrong work. I recommend checking it out.

Overall, the review is very good

Author Response

Thank you for your thoughtful feedback regarding our manuscript. We appreciate your comments and the opportunity to clarify some points related to our findings. New titles and paragraphs are marked in red.

In my opinion, a supromolecular approach is missing, namely association with surfactants and the introduction of PAMAM dendrimers as components of extracellular vesicles or synthetic micelles. But from the context of the review, it is clear that the authors focused on the most popular approaches.

R: While I appreciate the suggestion regarding a supramolecular approach, we focused on the most widely accepted methods in our article to specifically track the pathways of the modified dendrimer.

References 28 and 41 are duplicated.

  1. Zhu W, Okollie B, Bhujwalla ZM, Artemov D. PAMAM dendrimer-based contrast agents for MR imaging of Her-2/ neu
    receptors by a three-step pretargeting approach. Magn Reson Med. 2008 Apr 26;59(4):679–85.
  2. Zhu W, Okollie B, Bhujwalla ZM, Artemov D. PAMAM dendrimer-based contrast agents for MR imaging of Her-2/ neu
    receptors by a three-step pretargeting approach. Magn Reson Med. 2008 Apr 26;59(4):679–85.

R: Duplicate references resolved.

In my opinion, it would be a great omission not to mention the work of Jeanette C. Roberts, especially Preliminary biological evaluation of polyamidoamine (PAMAM) StarburstTM dendrimers (https://onlinelibrary.wiley.com/doi/10.1002/(SICI)1097-4636(199601)30:1%3C53::AID-JBM8%3E3.0.CO;2-Q).

R: The study was included between lines 180 and 192.

Link 63, which is the second one (conjugates carrying chemotherapeutics such as docetaxel (63).) leads to the wrong work. I recommend checking it out.

R: References were checked.

Round 2

Reviewer 1 Report (Previous Reviewer 2)

Comments and Suggestions for Authors

The author made revisions according to the comments of the reviewers. Also, the author made a point-to-point response to the comments. This manuscript can be accepted for publication in the current form.

This manuscript is a resubmission of an earlier submission. The following is a list of the peer review reports and author responses from that submission.

Round 1

Reviewer 1 Report

Comments and Suggestions for Authors

The review manuscript entitled “Navigating the cellular labyrinth: the journey of PAMAM dendrimers” written by Alamos-Musre and coworkers, examines different cell internalization pathways of functionalized PAMAM dendrimers. The interaction between the dendrimer charged surface and the cell membrane and mitochondria are crucial factors for evaluating therapeutic efficiency of PAMAM dendrimers as drug nanocarriers. The manuscript also discusses the toxicity of dendrimers to cells. While the topic presented in the review manuscript is of interest to the readership of Pharmaceutics, it fails to meet the journal’s quality requirements. The manuscript suffers from major inaccuracies that demerit its quality, below are major issues that need to be addressed.

Major issues.

In section “2.1 Poly(amidoamine) (PAMAM) dendrimers”, the authors discuss and introduce PAMAM dendrimers, even though these were already presented in the introduction section. Additionally, Figure 1 should be placed in the section where PAMAM dendrimers are mentioned for the first time.

In subsection “2.1.1 PAMAM dendrimer functionalization”, dendrimer functionalization is referred to structural or chemical modification, rather than drug interaction and transport mechanisms with dendrimers. Can the authors comment on this issue.

In subsection “2.3. Using PAMAM cytotoxicity as a potential biomedical use”, it is not clear how cytotoxicity of PAMAM dendrimers should be a therapeutic advantage as stated in the section title. The examples presented in this subsection focus on PAMAM dendrimers as carrier molecules, where the therapeutic agent is internalized and transported by PAMAM dendrimer. Can the authors elaborate on that issue.

In the introduction section, line 79 it is stated “The dendrimer core consists of one or more atoms, often carbon, nitrogen, or metals such as gold and silver, which provide the foundational framework for dendrimer synthesis.”  This statement is not accurate, dendrimer synthesis is typically performed by divergent or convergent methodologies. In divergent approach the synthesis starts at the core of the dendrimer, while in the convergent approach the branches are synthesized from the periphery towards the core.

It is highly recommended to avoid repetition of a topic in different sections or parts of the text. This is found line 68 and lines 72–78 where a general description of dendrimers is discussed twice. Dendrimer generation is defined twice, in lines 88 and 195. Dendrimer functionalization is mentioned in subsection 2.1.1 and in discussion section, line 585. Nucleotide and protein delivery in gene therapy using PAMAM dendrimers appear in the corresponding subsections 2.2.1. and 2.2.2. and in paragraphs of lines 606–616 and lines 617–628. Additionally, the PEG functionalization is also described in different sections of the manuscript (e.g. lines 679, 257, 378, 531)

A subsection of cytotoxicity of PAMAM dendrimers should be created, since this topic is described in detail in section “2.2.2. Mitochondria-Targeted PAMAM Dendrimers” including Figure 3, lines 410–458 and in lines 673–682. This is important since the conclusions are based greatly on cytoxicity of dendrimers.

It is highly recommended to include more Figures and schemes; only three figures were included it is not sufficient. For instance, in the introduction section a general scheme for a dendrimer with its structural components should be included.

All Figures must include a legal notice indicating permission to use the image, along with the precise reference from where it was taken.

A serious issue is the mismatch between references and what is stated in the text. This is of utmost significance, since otherwise the statements lack scientific foundation. Some examples are for instance, references 56, 60, and 64 are not about dendrimer macropinocytocis. References 28, 46 and 50 are not related to passive diffusion of PAMAM dendrimers of G4 (lines 334, and 342). Reference 67 is not related to G2 PAMAM dendrimer with and additional disulfide-linked outer layer (line 376). Reference 68 is not related to PEGylated PAMAM dendrimer (line 385).  References 76 and 79 do not deal with PAMAM dendrimer toxicity (line 433). Computational studies of nanocarriers are not studied in reference 97 (line 506). In line 556, reference 58 should be placed instead of reference 104. Reference 104 is not related to PAMAM dendrimers and Alzheimer’s disease (paragraph in line 566).

There are parts where citation is absent or proper citation is needed, for instance

  • line 546: “As a result, numerous studies have sought to functionalize dendrimers with targeting moieties to enhance specificity”
  • in line 301, references 44, 56–58 have no relation with the escape mechanism of cationic dendrimers from endosomes. Reference from Hamblin and coworkers (Appl. Mater. Today, 2018, 12, 177-190, DOI:10.1016/j.apmt.2018.05.002) should be included here instead.
  • line 312, “despite its advantages, this endosomal escape mechanism remains a subject of ongoing investigation”, references that support ongoing debate are missing.
  • line 368, “This interaction, mediated by the phosphate groups of DNA and the cationic charges of PAMAM dendrimers, …”, references 22 and 66 do not deal precisely with interaction between phosphate groups of PAMAM dendrimers and DNA.

Minor details

  1. A review manuscript does not include a results section; this should be changed.
  2. Acronyms must be described the first time they are mentioned, and repetition of acronym definition should be avoided.
  3. All the names of journals should be abbreviated using CASSI style. Format errors in the list of references must be corrected, for references 64, 79, 84, 85, 95 and 117. Additionally, references 68 and 81 are repeated, this should be corrected too. Except when it is strictly necessary, references should refer to works written in English language, see for instance references 4, 64, 76, 79, 94, 95.
  4. The numbering of references seems to lack of logic sequence, for instance reference clusters such as [28, 46, 50] in lines 342 and 349, [44, 56, 60] in lines 330 and 334, [15, 28, 76, 77] in line 417 and [6, 8, 22, 31, 32, 75, 92, 98, 99] in line 509 should be avoided. In addition, reference 82 appears before reference 80 in line 436.

Overall, the manuscript is challenging to read due to unclear organization. The examples provided could be discussed in more detail. Regarding manuscript quality, there is plenty of room for improvement.

Comments on the Quality of English Language

The reviewer recommends an English style revision by a native speaker. This manuscript needs to be rewritten with special attention on the organization of topics and ensuring references are accurately matched. 

Author Response

Revisor 1:

Major issues.

In section “2.1 Poly(amidoamine) (PAMAM) dendrimers”, the authors discuss and introduce PAMAM dendrimers, even though these were already presented in the introduction section. Additionally, Figure 1 should be placed in the section where PAMAM dendrimers are mentioned for the first time.

R: done as suggested

In subsection “2.1.1 PAMAM dendrimer functionalization”, dendrimer functionalization is referred to structural or chemical modification, rather than drug interaction and transport mechanisms with dendrimers. Can the authors comment on this issue.

R: comments done as suggested

In subsection “2.3. Using PAMAM cytotoxicity as a potential biomedical use”, it is not clear how cytotoxicity of PAMAM dendrimers should be a therapeutic advantage as stated in the section title. The examples presented in this subsection focus on PAMAM dendrimers as carrier molecules, where the therapeutic agent is internalized and transported by PAMAM dendrimer. Can the authors elaborate on that issue.

R: this subsection was very confused, so we decided to remove it.

In the introduction section, line 79 it is stated “The dendrimer core consists of one or more atoms, often carbon, nitrogen, or metals such as gold and silver, which provide the foundational framework for dendrimer synthesis.”  This statement is not accurate, dendrimer synthesis is typically performed by divergent or convergent methodologies. In divergent approach the synthesis starts at the core of the dendrimer, while in the convergent approach the branches are synthesized from the periphery towards the core.

R: the required information was added along with the removal of the wrong information.

It is highly recommended to avoid repetition of a topic in different sections or parts of the text. This is found line 68 and lines 72–78 where a general description of dendrimers is discussed twice. Dendrimer generation is defined twice, in lines 88 and 195. Dendrimer functionalization is mentioned in subsection 2.1.1 and in discussion section, line 585. Nucleotide and protein delivery in gene therapy using PAMAM dendrimers appear in the corresponding subsections 2.2.1. and 2.2.2. and in paragraphs of lines 606–616 and lines 617–628. Additionally, the PEG functionalization is also described in different sections of the manuscript (e.g. lines 679, 257, 378, 531).

R: comments done as suggested

A subsection of cytotoxicity of PAMAM dendrimers should be created, since this topic is described in detail in section “2.2.2. Mitochondria-Targeted PAMAM Dendrimers” including Figure 3, lines 410–458 and in lines 673–682. This is important since the conclusions are based greatly on cytoxicity of dendrimers.

R: this information was confused, so we decided to remove it

It is highly recommended to include more Figures and schemes; only three figures were included it is not sufficient. For instance, in the introduction section a general scheme for a dendrimer with its structural components should be included.

R: figure of a dendrimer is included in the introduction

All Figures must include a legal notice indicating permission to use the image, along with the precise reference from where it was taken.

R: Figures were made by us using Biorender premium version.

A serious issue is the mismatch between references and what is stated in the text. This is of utmost significance, since otherwise the statements lack scientific foundation. Some examples are for instance, references 56, 60, and 64 are not about dendrimer macropinocytocis. References 28, 46 and 50 are not related to passive diffusion of PAMAM dendrimers of G4 (lines 334, and 342). Reference 67 is not related to G2 PAMAM dendrimer with and additional disulfide-linked outer layer (line 376). Reference 68 is not related to PEGylated PAMAM dendrimer (line 385).  References 76 and 79 do not deal with PAMAM dendrimer toxicity (line 433). Computational studies of nanocarriers are not studied in reference 97 (line 506). In line 556, reference 58 should be placed instead of reference 104. Reference 104 is not related to PAMAM dendrimers and Alzheimer’s disease (paragraph in line 566).

R: References were reviewed and replaced as necessary.

There are parts where citation is absent or proper citation is needed, for instance

  • line 546: “As a result, numerous studies have sought to functionalize dendrimers with targeting moieties to enhance specificity”
  • in line 301, references 44, 56–58 have no relation with the escape mechanism of cationic dendrimers from endosomes. Reference from Hamblin and coworkers (Appl. Mater. Today, 2018, 12, 177-190, DOI:10.1016/j.apmt.2018.05.002) should be included here instead.
  • line 312, “despite its advantages, this endosomal escape mechanism remains a subject of ongoing investigation”, references that support ongoing debate are missing.
  • line 368, “This interaction, mediated by the phosphate groups of DNA and the cationic charges of PAMAM dendrimers, …”, references 22 and 66 do not deal precisely with interaction between phosphate groups of PAMAM dendrimers and DNA.

R: The missing references were added and those that needed replacing were substituted.

Minor details

  1. A review manuscript does not include a results section; this should be changed.

R: According to the structure provided by the journal, the results section was required. Any modification to this structure must be indicated by the journal's editorial board.

  1. Acronyms must be described the first time they are mentioned, and repetition of acronym definition should be avoided.

R: comments done as suggested

  1. All the names of journals should be abbreviated using CASSI style. Format errors in the list of references must be corrected, for references 64, 79, 84, 85, 95 and 117. Additionally, references 68 and 81 are repeated, this should be corrected too. Except when it is strictly necessary, references should refer to works written in English language, see for instance references 4, 64, 76, 79, 94, 95.

R: the format of the bibliography section was standardized. Duplicated bibliographies were corrected. Spanish references were replaced.

  1. The numbering of references seems to lack of logic sequence, for instance reference clusters such as [28, 46, 50] in lines 342 and 349, [44, 56, 60] in lines 330 and 334, [15, 28, 76, 77] in line 417 and [6, 8, 22, 31, 32, 75, 92, 98, 99] in line 509 should be avoided. In addition, reference 82 appears before reference 80 in line 436.

R: The citation format in the text was corrected and standardized.

Reviewer 2 Report

Comments and Suggestions for Authors

This review focuses on the cellular internalization pathways, cytotoxicity, and functionalization strategies of PAMAM dendrimers for biomedical applications, providing a favorable background introduction and support for clinical translational advances of PAMAM dendrimers.However the amount of content in the article is low and needs to be supplemented and some of the conclusions need to be supplemented with more examples.
1. Add a description of the synthetic methodology of PAMAM dendrimers, including the selection of core molecules, control conditions of branching units, etc., and mention the possible challenges of the synthesis process.
2. Add more examples of the application of PAMAM dendrimers in anticancer drug delivery.
3. Add the application of PAMAM dendrimers in vaccine delivery, especially how to improve the immunogenicity of vaccines through surface modification.
4. Add more studies on targeting ligands and imaging in the section on functionalized modifications.
5. Add the advantages and disadvantages of PAMAM dendrimers compared with other non-viral vectors in gene transfection.
6. Can more complete mechanistic studies or data be provided to support the conclusion that cationic PAMAM dendrimers are more cytotoxic than anionic or neutral PAMAM dendrimers?
7. Comparison of the effects of different cell types or cell membrane compositions on the internalization pathways of PAMAM dendrimers can be added.
8. It is mentioned in the paper that PEGylation modification can play a series of roles, are there any research papers reporting on it?Some examples can be added to confirm this.
9. The complexity of the metabolism and clearance pathways of PAMAM dendrimers in vivo is not mentioned in the text, does this affect their clinical translational potential?

Comments on the Quality of English Language

The English could be improved to more clearly express the research.

Author Response

However, the amount of content in the article is low and needs to be supplemented and some of the conclusions need to be supplemented with more examples.

  1. Add a description of the synthetic methodology of PAMAM dendrimers, including the selection of core molecules, control conditions of branching units, etc., and mention the possible challenges of the synthesis process.

R: requested information added.

  1. Add more examples of the application of PAMAM dendrimers in anticancer drug delivery.
    R: requested information added.

  1. Add the application of PAMAM dendrimers in vaccine delivery, especially how to improve the immunogenicity of vaccines through surface modification.
    R: requested information added.

  1. Add more studies on targeting ligands and imaging in the section on functionalized modifications.
    R: requested information added.

  1. Add the advantages and disadvantages of PAMAM dendrimers compared with other non-viral vectors in gene transfection.
    R: requested information added.

  1. Can more complete mechanistic studies or data be provided to support the conclusion that cationic PAMAM dendrimers are more cytotoxic than anionic or neutral PAMAM dendrimers?
    R: requested information added.

  1. Comparison of the effects of different cell types or cell membrane compositions on the internalization pathways of PAMAM dendrimers can be added.
    R: we did not find enough information to include this issue.

  1. It is mentioned in the paper that PEGylation modification can play a series of roles, are there any research papers reporting on it? Some examples can be added to confirm this.
    R: requested information added.
  2. The complexity of the metabolism and clearance pathways of PAMAM dendrimers in vivo is not mentioned in the text, does this affect their clinical translational potential?

R: requested information added.

Round 2

Reviewer 1 Report

Comments and Suggestions for Authors

Otero and coworkers presented a revised version of the manuscript “Navigating the cellular labyrinth: the journey of PAMAM dendrimers”. New sections such as: PEGylation (pp. 13–14), PAMAM dendrimers in vaccine delivery (pp.9–10) and some applications and advantages of PAMAM dendrimers on pages 15–16, have been added to the manuscript.  

The reviewer considers the authors failed to give a sufficient response to point-by-point comments and corrections. The order of sections in the whole manuscripts lacks pertinent logic, there are topics which are discussed in different parts of the manuscript and it seems the information is repeated, see for instance synthesis of PAMAM dendrimers, PEGylation of PAMAM dendrimers, among others. . As previously suggested by the reviewer, the manuscript section “Results” is not recommended in review articles. 

The synthesis of PAMAM dendrimers is described first in line 96 (p. 3) and in section 2.1 “PAMAM synthesis” line 213 (p. 5); it is recommended to include this information together in one section. This section and other sections would be greatly enriched with schemes and figures, nevertheless the authors failed to comply with this comment. An inconsistency already mentioned by the reviewer was that section “2.1.2. PAMAM dendrimer functionalization”, contains mostly drug interaction with dendrimers and transport mechanisms rather than PAMAM structural functionalization. For instance, the examples contained in the manuscript on PAMAM PEGylation could be included in this section. The sentence “The dendrimer core consists of one or more atoms, often carbon, nitrogen, or metals such as gold and silver, which provide the foundational framework for dendrimer synthesis.”, is not accurate, as it has already been pointed out by the reviewer.

Structure of PAMAM dendrimers in Figure 1 is a general scheme, nor specific for PAMAM dendrimer. It was suggested by the reviewer to include a separate section on cytotoxicity of PAMAM dendrimers. How is this information confusing, as the authors replied. One of the three figures in total of the manuscript depicts the general toxic effects of these constructs. Finally, the “references” section has been revised and corrected as stated by the authors but there are still inconsistencies, some of them will be mentioned below. References 4 and 121 cite to sources in Spanish language. The Book chapter reference 94 is incomplete. Some style errors were found, for instance in the title of reference 98 has a typo error and the names of authors are written in capital letters in reference 167. An important detail to consider is to cite the correct sources, for example reference 150 does not correspond to Abd-El-Azi and coworkers.

The manuscript in its present form has insufficient quality to be published. The reviewer recommends resubmission.

Comments on the Quality of English Language

The manuscritpt new version compared with the previous one has not been revised in terms of English language quality, as recommended by the reviewer. 

Reviewer 2 Report

Comments and Suggestions for Authors

The author made revisions according to the comments of the reviewers. Also, the author made a point-to-point response to the comments. I have no further comment.